# Lifestyle and Sense of Coherence: A comparative analysis among university students in different areas of knowledge

**João Paulo Costa Braga**\*, **Eduardo Wolfgram, João Paulo Batista de Souza, Larissa Gabriele Fausto Silva, Yonel Estavien, Roberto de Almeida, Cezar Rangel Pestana**

Medical School, UNILA University, Foz do Iguaçu, Parana, Brazil

\* jpcbmed@gmail.com

**Data Availability Statement:** All data presented in the manuscript are fully available at https://datadryad.org/stash/share/_

## Abstract

### Background

The concept of health has undergone profound changes. Lifestyle Medicine consists of therapeutic approaches that focus on the prevention and treatment of diseases. It follows that the quality of life of university students directly affects their health and educational progress.

### Experimental methodology

Socioeconomic, lifestyle (LS), and Salutogenesis Theory/sense of coherence (SOC) questionnaires were administered to college students from three different areas. The results were analyzed for normality and homogeneity, followed by ANOVA variance analysis and Dunn and Tukey post hoc test for multiple comparisons. Spearman's correlation coefficient evaluated the correlation between lifestyle and sense of coherence; p values < 0.05 were considered statistically significant.

### Results

The correlation between LS and SOC was higher among males and higher among Medical and Human sciences students compared to Exact sciences. Medical students' scores were higher than Applied sciences and Human sciences students on the LS questionnaire. Exact science students' scores on the SOC questionnaire were higher than Human sciences students. In the LS areas related to alcohol intake, sleeping quality, and behavior, there were no differences between the areas. However, women scored better in the nutrition domain and alcohol intake. The SOC was also higher in men compared to women.

### Conclusion

The results obtained demonstrate in an unprecedented way in the literature that the correlation between the LS and SOC of college students varies according to gender and areas of knowledge, reflecting the importance of actions on improving students' quality of life and enabling better academic performance.

RlHZJP5BJckyC4EZQ2_Q-Rjtj39aMxSbe-3sLTDhko and https://doi.org/10.5061/dryad.bcc2fqzhd.

**Funding:** The Grant number: PRPPG n° 2020/096, awarded to CRP was given from Universidade Federal da Integração Latino Americana - Pro-Reitoria de Pesquisa e Pós-Graduação - https://portal.unila.edu.br/prppg. The funders had no role in study design, data collection and analysis, decision to publish, or preparation of the manuscript.

**Competing interests:** The authors have declared that no competing interests exist.

## Introduction

Health is described as "a state of complete physical, mental, and social well-being, not merely the absence of disease or infirmity" in the WHO Constitution from 1946. This was expanded by Lalonde (1974), Young (1982), and Canguilhem (1990) to include the interaction between physical, mental, social and environmental factors [1]. The significance of lifestyle studies has grown as a result. To highlight the complexity of various factors influencing health, a rainbow model of health determinants was developed [2]. Rice and Sara [3] proposed an update to this model that estimate developments that have occurred over the last three decades by the strengthening of the digital environment and It was also probably amplified during the moment of isolation imposed by the COVID19 pandemic.

From the 1990s, quality of life research has increased [4, 5]. Several studies have shown that healthy lifestyle choices like exercising, eating a balanced diet of fruits and vegetables, cutting back on drinking and smoking, and managing stress all contribute to wellbeing. Moreover, unhealthy lifestyle choices are the cause of two-thirds of diseases. An active, healthy lifestyle results in better living circumstances and lower medical costs [6].

A person's lifestyle is a collection of consistent behaviors that reflect their attitudes and circumstances. The six pillars of lifestyle medicine (LSM) are a primarily plant-based diet, regular exercise, getting enough sleep, managing stress, avoiding hazardous substances, and fostering healthy interpersonal relationships [7]. Beyond treating symptoms and preventing disease, LSM aims to boost wellbeing (salutogenesis) and give people the tools they need to lead healthy lives [8]. The lifestyle of undergraduate students has been measured worldwide using the Fantastic Lifestyle Questionnaire for many years, both before [9] and during the COVID-19 pandemic [10].

The central concept in Antonovsky's Salutogenesis Theory, which helps people deal with difficult and stressful circumstances better, is the Sense of Coherence (SOC) [11, 12]. Cognitive (understanding), behavioral (management), and motivational (meaning) are its three primary elements [13, 14]. This idea contends that a strong sense of coherence enhances motivation and purpose in one's perception of the world, leading to lower levels of stress and anxiety [13, 14]. According to research on the predictors of quality of life in medical students, time management is a protective factor in enhancing quality of life and a key to healthy balance in a variety of professional careers [15].

College students' quality of life is frequently impacted by a variety of factors, including family and health issues, financial worries, and career decisions [16]. It is intimately related to ideas about health, happiness, and lifestyle [17]. College students frequently have sleep problems and daytime tiredness, which can have a negative impact on their memory, learning, and general health [18]. It is also common to find a high burnout prevalence among medical students in first years of study [19].

According to Eriksson and Lindström [20], the sense of coherence (SOC) is consistently correlated with a higher quality of life and better health. According to studies, people with high SOC scores have healthier lifestyles, including more physical activity, less smoking and alcohol-related disorders, and better eating habits. They also report feeling better about their psychological well-being and work satisfaction. These results suggest that a strong SOC is necessary for keeping positive habits and can facilitate lifestyle changes [21, 22].

As reported by Antonovsky [11], the sense of coherence is predominantly developed during childhood and adolescence. A reciprocal longitudinal link between SOC and both self-reported or physician-assessed health in people over 33 years old supported the idea that SOC is improved with time [23].

A study of nearly 500 students at the Federal University of Juiz de Fora, Brazil showed the human sciences area had best averages for quality of life [24]. Another study of medical & STEM (science, technology, engineering, and mathematics) freshmen students at the University of Lübeck, Germany showed medical students had higher scores & less depression (96.6% vs 92.2%), as well as a higher percentage exercising regularly & engaging in musical activity [25]. But a study on mental disorders among health students from the University of Pernambuco, Brazil found the highest prevalence in medical students (42.6%), followed by dental students (33.3%), nursing (31.8%) & physical education (25%) [26]. This divergence between studies suggests the data's real statistical significance is uncertain, probably due to different countries, methodologies, assessment tools, & lack of further research.

Higher emotional intelligence is linked to better academic performance among final-year medical students as well as among nursing students, according to a study conducted by Ranasinghe P. et al. [27]. This study examined the relationship between emotional intelligence, perceived stress, and academic outcomes among a large cohort of medical students from three distinct years of the undergraduate medical curriculum. Slimmen S. et al. demonstrated that students who experience higher levels of stress are at a higher risk of suffering from mental health challenges [28].

This research is intended to investigate the hypothesis that college students' lifestyles (LS) and senses of coherence (SOC) in a Brazilian university are positively correlated. The correlation between these variables suggests that students' psychological well-being declines in the final years of their coursework, most likely as a result of stress, exhaustion, and anxiety [29], although emotional intelligence among medical students in Southern Asia, significantly improved over 5-years of follow-up, as shown by Ranasinghe P et al. in a study with 170 students [30]. We chose to investigate students mostly in the middle of their courses. While several research have found a correlation between LS and SOC, the objective of this study is to examine how SOC and LS are related across genders and academic subjects, such as the exact sciences, biological sciences, and humanities. To the best of our knowledge, this research is the first of its type to look at this association specifically in terms of gender and academic discipline. In the databases searched, there was no existing literature correlating these two factors together.

## Materials and methods

### Participants, inclusion criteria and COVID pandemic

The participants are students of both genders, enrolled in courses in the Health area (HA), Applied sciences area (ASA) and Human sciences (HM). In the Health area, the questionnaires were applied to medical students, Applied sciences for Engineering, and Human sciences for History, totaling 156 students. After authorization from the professor in charge, the participants were recruited in their online classes, where the research was presented and they were invited to participate.

The inclusion criterion was the students' active enrollment in the courses, excluding students who were in their first and last semester as a way to select students who were already far from their family customs and the stress overload due to the proximity of the course completion.

The questionnaire was applied online through the Google Forms platform between December 2020 and August 2021 due to the restrictive measures of the COVID pandemic. Given this context, the questionnaire responses and their results, may have been influenced by the uncertainties and weaknesses experienced at the time of completing the online questionnaire, as well as by isolation restrictions.

## Data and study size

The participants' data were collected through three questionnaires, specified below and authors had NO ACCESS to information that could identify individual participants during or after data collection.

We used a number of approaches to counteract potential biases in our study. To assure a sample of the population that was representative of the public, we chose participants from a variety of backgrounds, including students from various years, genders, and periods. Each student answered to their questions independently, without access to or communication with any other student. Furthermore, we made efforts to protect confidentiality throughout the data collecting and analysis procedure. Because we were unable to access the names of the students who responded to the questionnaires, any potential biases resulting from prior preconceptions about particular people were lessened.

We randomly selected about 50 among 200 students of each study area. Questionnaires were applied individually to avoid potential bias.

## Socio-economic questionnaire

Tool for collecting information on specific aspects of life, and socio-economic conditions, forming the basis for a socio-economic assessment of the participants [31]. Before completing this questionnaire, the students were informed about the research and its objectives by reading the Free and Informed Consent Form. Soon after, the socio-economic questionnaire was applied, with personal information, such as gender, age, origin and family arrangements.

## Lifestyle questionnaire

The Fantastic Lifestyle Questionnaire, created by Wilson and Ciliska in 1984, is a validated tool for evaluating lifestyle behaviors such as physical exercise, interpersonal connections, sleep, nutrition, introspection, and job satisfaction. It is a simple and objective way to cover a variety of topics that can have an impact on one's health, such as responsible sex, the usage of seatbelts, and alcohol and drug use. In order to give a complete picture of a person's lifestyle, the questionnaire specifically examines behavior during the previous month [32].

The word "Fantastic" is an acronym, which represents the first letters of the name of the 9 analyzed domains:

F = *Family and friends*;
A = *Activity* (physical activity);
N = *Nutrition*;
T = *Tobacco & toxics*;
A = *Alcohol*;
S = *Sleep, seatbelts, stress, safe sex*
T = *Type of behavior* (behavior type; behavior pattern A or B);
I = *Insight* (introspection);
C = *Career* (job; job satisfaction).

The questionnaire comprises 25 questions, with five alternative answers, arranged in the form of a Likert scale, ordered in columns. The alternative on the left always has the lowest value or has the least relation to a healthy lifestyle. The score for each question is divided as follows: 0 points for the first column, 1 for the second, 2 for the third, 3 for the fourth, and 4 for the fifth column. Questions with only two alternatives are scored: zero for the first column and 4 points for the last column. The sum of all the points gives a total score that classifies individuals into five categories, which are: "Excellent" (85 to 100 points), "Very good" (70 to 84 points), "Good" (55 to 69 points), "Fair" (35 to 54 points), and "Needs improvement" (0 to 34 points).

The lower the score, the greater is their need for change; being desirable that individuals achieve a score above 55 [32].

## Sense of coherence questionnaire

A closed and systematized questionnaire was applied to evaluate the sense of coherence. This tool was proposed and created in 1979 by Antonovsky, and the participants were rated on a ten-point scale, classifying as strong, moderate, and weak SOC. The content of the questionnaire was based on the SOC components (understanding, management, and meaning) and stimuli experienced by people divided on modality (instrumental, cognitive, and affective), source (internal, external, or both), nature of the demand imposed by the stimulus (concrete, diffuse, or abstract), and the stimulus reference time (past, present, or future). A total of 29 questions were obtained after this process, with each question including these five facets. The author entitled the instrument an Orientation to life questionnaire but has been referred to in the literature as Antonovsky's Sense of Coherence questionnaire (ASCQ) [13].

Responses to the items were obtained using a seven-point scale, with the values ranging from one (1) to seven (7), in which the value one (1) represents the weakest SOC and seven (7) the strongest. Of the 29 items, 13 were answered on a reverse scale of values, i.e., higher values indicate lower SOC (questions 1, 4, 5, 6, 7, 11, 13, 14, 16, 20, 23, 25, and 27). High values always mean a strong Sense of Coherence. The possible scoring ranges from 29 to 203 [13].

## Data gathering and ethics statement

The researchers invited the students to answer an online form—through *Google Forms* virtual platform—containing the questionnaires: sociodemographic information, FANTASTIC questionnaire on Lifestyle, a questionnaire on Sense of Coherence and the Free and Informed Consent Form. The researchers clearly explained the research objectives and collection procedures on the home page, and the participants consent the use of the information through the written online Consent Form.

The data gathered in the online form were transferred to a spreadsheet in *Microsoft Excel*. The results were filtered, classified, and treated in order to be in line with the desired statistical analysis and could feed the statistical programs used.

The Ethics Committee of the Centro Universitário Dinâmica das Cataratas approved this study by consolidated approval under number: 4.157.260 of 07/15/2020.

## Statistical analysis

The statistical analyses were performed by the JASP statistical *software*, and part of the graphics by the SPSS *software*. First, the researchers submitted the results to normality (Shapiro Wilk) and homogeneity (Levene test) analysis. Next, the normal homogeneous data were submitted to the ANOVA analysis of variance and Kruskal Wallis non-parametric test, followed by Dunn's post hoc test of multiple comparisons and Tukey's correction.

Spearman's correlation coefficient evaluated the correlation between Lifestyle and Sense of Coherence by determining the value of R. Values of $p < 0.05$ were considered statistically significant.

The normality of data was checked by the Shapiro-Wilk test, and since the distribution was not normal, analyses were performed as described below:

a. Descriptive results are presented by the median and interquartile ranges.

b. The comparisons between the study variables in HA, ESA and HM (age, BMI, lifestyle, sense of coherence and domains of the questionnaires) and by gender (Lifestyle and Sense of Coherence) were performed by the Mann-Whitney test.

c. The domains of the questionnaires in each group (HA, ESA, and HM) were compared by analyzing repeated measures, *Friedman* test, and the *Post Hoc* by *Dunn*'s multiple comparisons test.

d. The comparisons between lifestyle and Sense of Coherence among students in each of the selected courses were performed by analysis of variance, *Kruskal-Wallis* non-parametric test and *Post Hoc* by *Dunn*'s multiple comparisons test.

e. The correlations between the profile of lifestyle and sense of coherence of students in each area of knowledge and by gender were performed by *Spearman*'s correlation coefficient.

The significance index adopted in all analyses was 5% ($p \leq 0.05$).

## Results

### Student's socio-demographic profile

Among the study participants, 59 were Medical students, 42 Human sciences students, and 55 Applied sciences students, totaling 156. The overall average age was 25 years. There was a prevalence of 51.92% of female students in all courses, especially in the area of Human sciences, with a percentage of 57.14% of women. The overall median BMI was 23.39, with a minimum of 15.57 and a maximum of 46.38. Regarding self-declaration of ethnicity, 51.92% of the interviewed students declared themselves white, 8.97% black, 30% brown/mulatto/caboclo, 1.92% oriental yellow, and 7.05% indigenous.

Brazilian students represented 62.82% of the participants, besides the presence of other Latin American nationalities, described in alphabetical order as follows: Argentina, Bolivia, Colombia, Costa Rica, Cuba, Dominican Republic, Ecuador, Guatemala, Haiti, Honduras, Paraguay, Peru, El Salvador, Venezuela and Pakistan as a non-Latin American country. In addition, 93.59% of the participants are from the urban area. About their current residence, 86.54% live with someone, 55.77% live in a single-story house and 73.72% rented. 97% Declared not having another university course, and 23.72% are working an average of 5.76 hours a day. The students' economic profile distribution shows that 62.18% receive financial aid from parents and relatives.

### Overall score of questionnaires

Scores on the Fantastic questionnaire indicated that Medical students have a better Lifestyle, rated as 'very good' compared to students in the Applied sciences and Human sciences, who scored a Lifestyle rating of 'good' (Fig 1). The scores obtained for Lifestyle were statistically higher for Medical students in comparison to the Applied sciences (71.136 +- 11.276 vs 65.982 +- 13.144, p = 0.004) and to the Human sciences students (71.136 +- 11. 276 vs 66.405 +- 10.411, p = 0.011), being classified in the FANTASTIC questionnaire score as very good (70–84—Provides adequate influence on health) and good (55–69—Provides many health benefits). As for the Sense of Coherence, Medical students also presented higher scores, but all groups obtained the median considered as a 'high Sense of Coherence' (Fig 2). As for the Sense of Coherence score, the area of Applied sciences was statistically higher than Human sciences (78.418 +- 21.235 vs 70.786 +- 17.728, p = 0.039), as presented in Fig 2. There was no statistically significant difference between Applied sciences and Medicine (78.418 +- 21.235 vs 74.085 +- 17.037, p = 0.339) or between Human sciences and medicine (70.786 +- 17.728 vs 74.085 +-

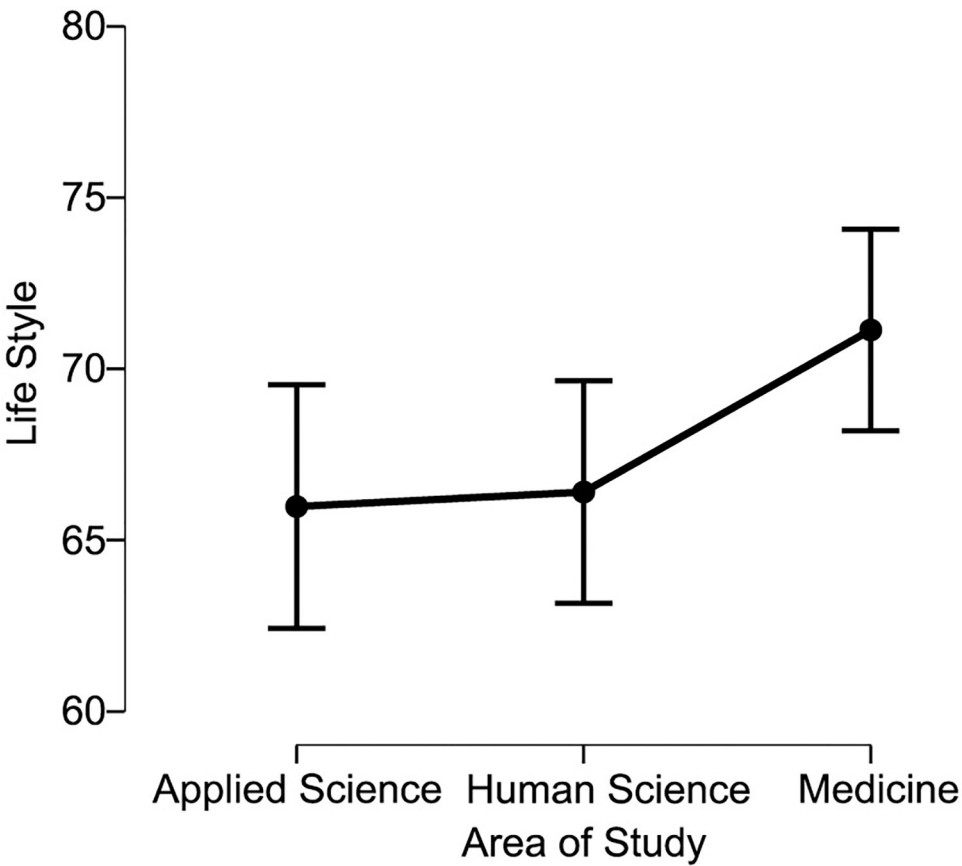

**Fig 1. Total lifestyle scores regarding each study area.** *p<0.05 were considered statistically significant.

17.037, p = 0.08) and other areas of knowledge. The classification is stratified into five levels of behavior regarding the quality of life, from 0 to 34: Needs improvement; 35 to 54: Fair; 55 to 69: Good; 70 to 84: Very good and 85 to 100: Excellent [33]. There is no specific methodology to categorize the SOC score, however, some authors have stipulated a score to better identify the levels of the sense of coherence. Thus, it is understood that a score from 13 to 38: low, between 39 and 65: moderate, and above 65: high [34].

## Comparison of both questionnaires' domains across study areas

The comparative results among the nine domains of the FANTASTIC questionnaire showed significant differences among the Health, Applied sciences, and Human sciences areas. Medical students showed statistically better values in the Family (p = 0.014) and Physical Activity (p = 0.012) domains when compared to the Applied science area. There was no significant difference between Human sciences and Applied sciences/Medicine. Regarding the Nutrition domain, there was a statistically significant difference between the Human sciences and Applied sciences areas (p = 0.044), with Applied sciences scoring better than Human sciences. There was also a statistically significant difference when comparing Medicine and Human sciences (p = 0.005), with Medicine scoring better. In the tobacco and drug use domain, Medicine scored significantly better than Applied sciences (p = 0.048).

Medical students showed higher results in the Introspection domain than Applied Sciences students (p = 0.036). This study also showed that career satisfaction was higher in medical

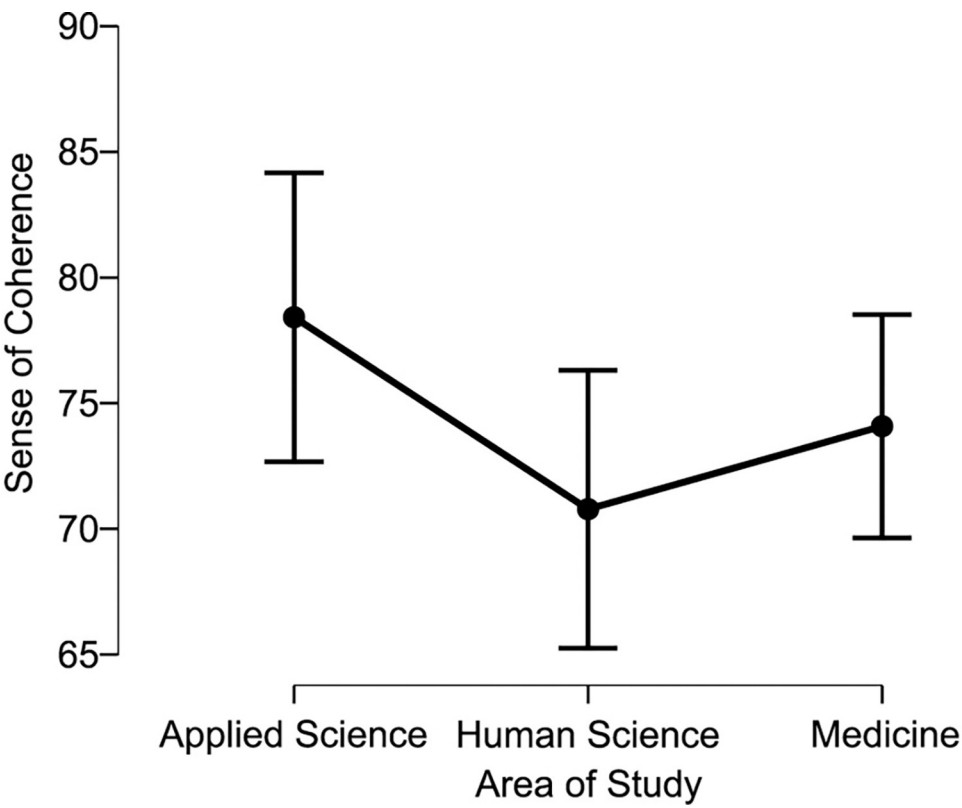

**Fig 2. Total sense of coherence scores regarding each study area.** *p<0.05 were considered statistically significant.

students when compared to applied sciences students (p = 2.443e -4) and also in comparison to Human sciences students (p = 0.029). As for the domain's pertaining to alcohol, sleep and behavior, there were no significant differences between the areas (Fig 3).

Regarding the Sense of Coherence questionnaire, Fig 4 shows that the Medical course presented statistically superior scores to the Human sciences course in the comprehension variable (p = 0.028). However, when analyzing the management domain, Applied sciences was statistically superior to Human sciences (p = 0.049), and in the meaning variable, no statistically relevant differences were observed among the three areas.

Compared to Applied Sciences, Fig 5 shows a more positive correlation between Sense of Coherence and higher Lifestyle in Medicine and Human Sciences, suggesting that students from these fields may lead a healthier lifestyle (see Fig 5). According to Fig 6, Sense of Coherence correlates more strongly with a health-promoting Lifestyle in males than females. In addition, there is a correlation between men who have a stronger sense of coherence and a healthier lifestyle, while women with a weaker sense of coherence are less likely to be healthy.

## Gender total score comparison on both questionnaires

Comparisons of gender-based total scores of the lifestyle (Fig 7) and Sense of Coherence domains (Fig 8).

Comparisons of domains based on gender show that men scored higher in comprehension (p < 0.003) and management (p < 0.001), but with no statistically significant differences (Fig 8).

The study shows that women have a healthier pattern of nutrition (women 7,852 +- 2,784 vs men 6,986 +- 2,869, p = 0.032) and lower alcohol consumption (women 10,654 +- 1,526 vs men 9,919 +- 2,235, p = 0.047).

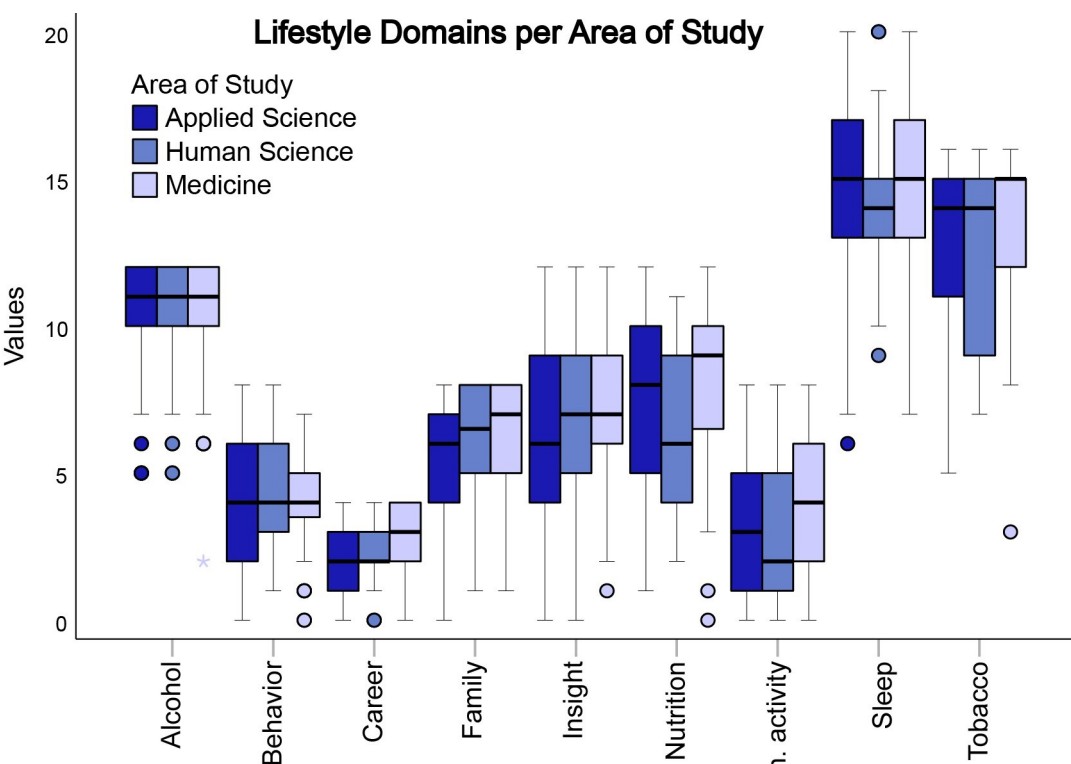

**Fig 3. Distribution of domains in the Applied sciences (blue), Human sciences (green), and Medicine (brown) areas, based on the Fantastic Lifestyle Questionnaire p < 0.05.**

## Discussion

The results establish a positive correlation between Sense of Coherence and lifestyle among the surveyed students. In this study, men had a higher overall score in the Sense of Coherence, more specifically in the domains of understanding and management, which several factors can partially explain. Among them, differences in the incidence of mental illness and resilience profile according to gender, as described by Almalik et al. [35], which pointed out higher levels of anxiety in women (78.3% versus 21.7%). Mazurek Melnyk et al. [36] also demonstrated a higher rate of depression in women, and Kötter T et al. showed that the two best predictors for enrolling in a health promotion course given in a medical course were being female and having more anxiety [37]. It is worth noting that all the mentioned research present differences in methodology and groups analyzed.

In the nutrition domain, women are more concerned about a good diet. Similar results were observed by Arroyo Izaga et al. [38], as well as Safarino et al. [39]. Women reported a preference for pre-prepared food or for purchasing ready-made food in contrast to men, and women tend to follow a diet with less meat, according to research by Murillo-Llorente MT et al. on the validity and reliability of the FANTASTIC Questionnaire for Nutritional and Lifestyle Studies in University Students [40]. The study area with the lowest score in the nutrition domain was Human sciences, which is different from what was observed in other studies in which Medicine obtained the worst results [33].

Among the interviewed students, 35% were overweight (BMI > 25), and Applied sciences and Human sciences students scored low in the physical activity domain. This percentage is higher than in other studies. For example, a study by Silva et al. [41], with first-year students

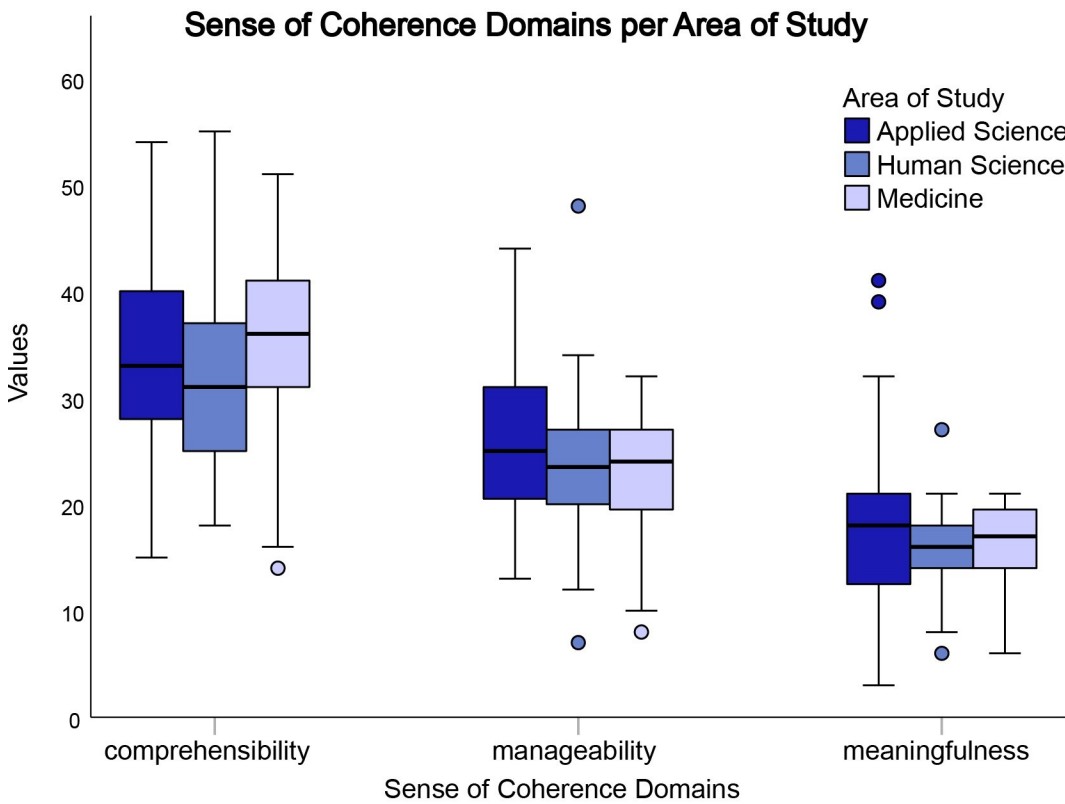

**Fig 4. Distribution of domains in the Applied sciences (blue), Human sciences (green), and Medicine (brown) areas, based on the Sense of Coherence p < 0.05.**

showed that 16% of the students analyzed were overweight, which emphasizes the importance of physical activity for disease prevention among college students. In addition, family plays a key role in the lives of college students through the triad of "Confidence, ability and reward", "Desire for social mobility", and " University structures and income" [42]. In our study, the family domain had a high average in all areas of the study (> 7pts), possibly influencing the average overall quality of life score higher than 65 pts considered as GOOD.

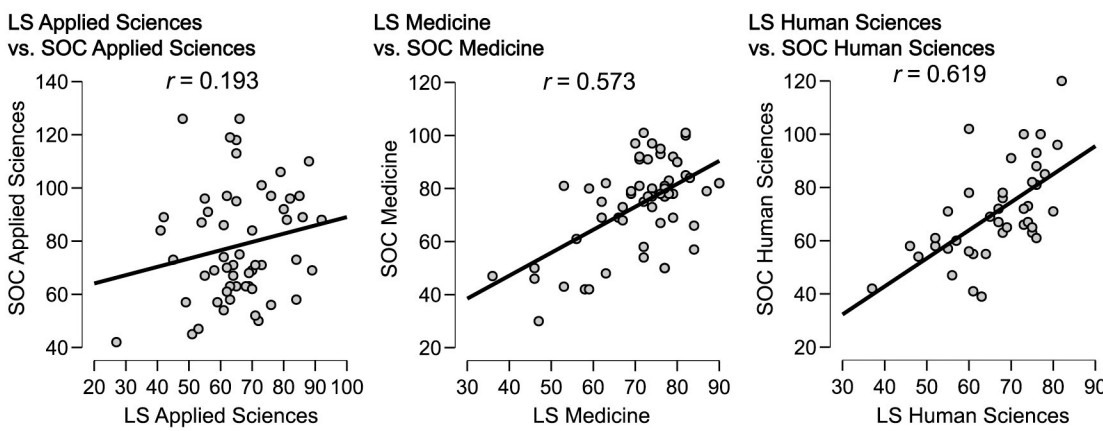

**Fig 5. Correlation between Lifestyle and Sense of Coherence by area of knowledge.**

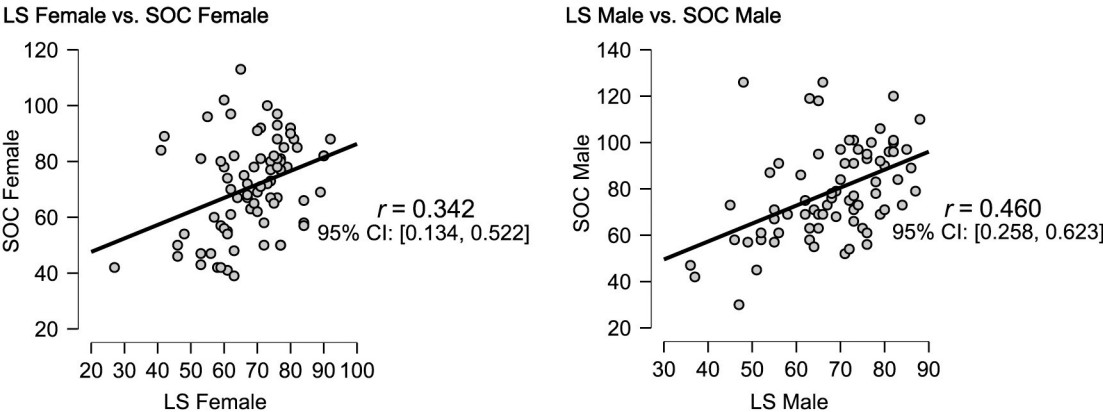

**Fig 6. Correlation between Lifestyle and Sense of Coherence by gender.**

Medical students performed better on the Lifestyle Questionnaire than students in other courses despite the course's considerable stress [43]. Possibly due to modules on salutogenic lifestyle included in the curriculum, UNILA medical students got the highest scores in the Introspection domain. This outcome is comparable to one from a German study that looked at the effects of psychosocial stress in STEM and medical students. In contrast to medical students, the study indicated that a substantial and increasing percentage of STEM students are at

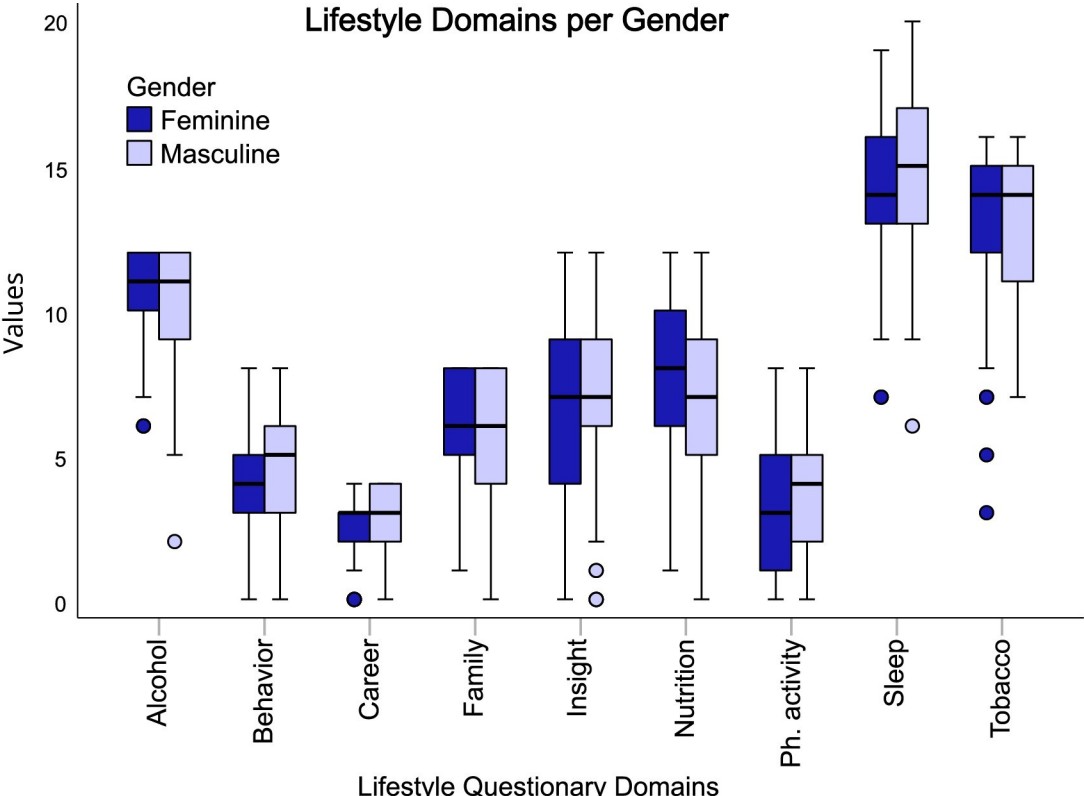

**Fig 7. Box plot representing the total score of each domain of the Fantastic questionnaire divided by male and female genders.**

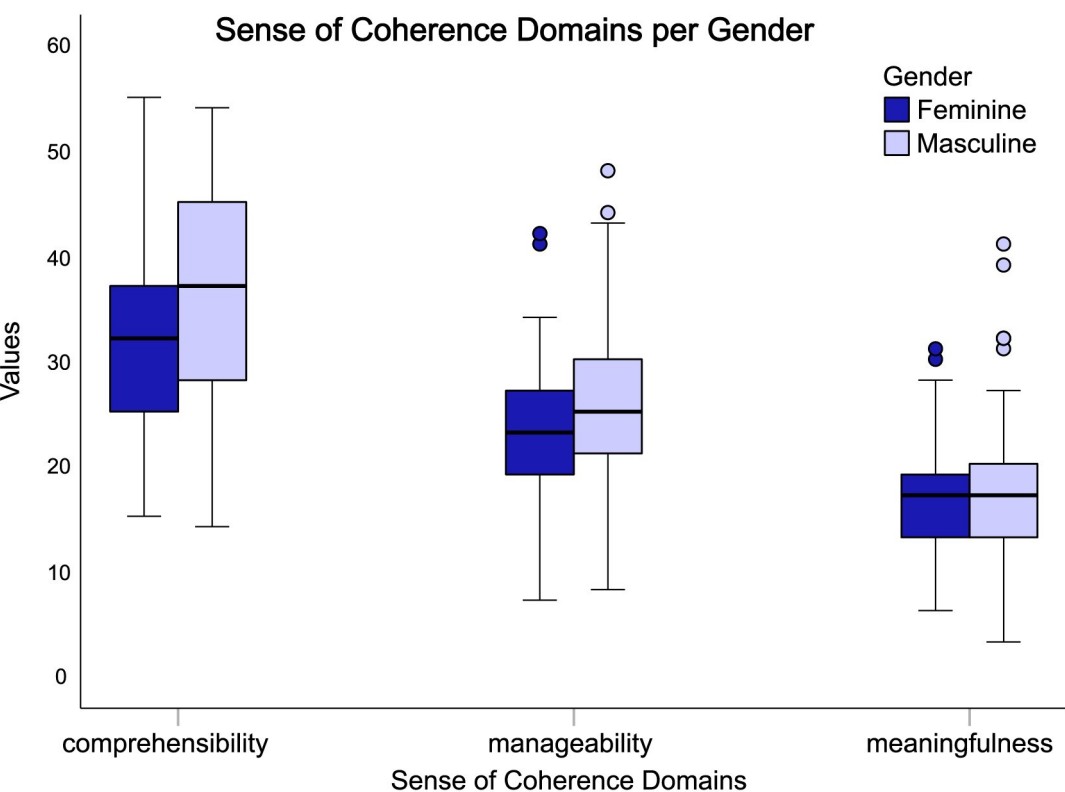

**Fig 8. Box plot representing the total score of each Sense of Coherence domain, divided by male and female genders.**

risk for burnout and that a smaller fraction display healthy study-related behavior and experience patterns. This difference can be probably explained, by the theory that stress may not be so much pronounced in medical education, which might have an impact on the outcomes [44]. However, in our study dietary habits from medical students were not as healthy, with high fast food, soft drinks and alcohol consumption and more sedentary lifestyle. Similar to earlier research, smoking was the area where educational intervention was most needed [45].

Regarding gender, Abreu et al. [46], observed that men are more likely to consume alcohol in excess compared to women, corroborating the results of this research. However, in others, alcohol consumption among women was higher than men [47].

Regarding the Sense of Coherence, we saw in this study the existence of a positive relationship between it and Lifestyle, demonstrated by both average scores considered good. Da Silva, [34], analyzed different moments of ambiguity, anxiety, and expectations among students, indicating that students have difficulty with understanding and managing the resources available to deal with adverse circumstances faced during their journey, with an increasingly high rate of depression and suicide. The positive association between the sense of coherence and quality of life can contribute to better academic performance and professional profile, and a greater sense of coherence was associated with less somatic stress symptom manifestation., as shown by Sójka A et al. in their study with 324 medical student [48]. Pantuza et al. [14], demonstrated that college students with higher SOC had more confidence to speak in public and experienced less labored breathing. Thus, being able to recognize stressful situations, mobilize resources to promote effective coping, and find solutions to resolve situations and difficulties, in order to have a better quality of lifestyle and health. Jurczyszyn A. and Zdziarski K. [49], showed in a study on sense of coherence among 145 Polish university students a slightly lower

sense of coherence among students studying medicine compared to a group from non-medical universities, in contrast to our study, which found no statistically significant difference between medicine and the other areas searched.

## Conclusion

The positive correlation between Lifestyle and Sense of Coherence in college students varies by gender and areas of knowledge. Nevertheless, the results obtained may be helpful for the academic community to provide students, professors, and professionals with data on Lifestyle and Sense of Coherence of UNILA's college students and thus foster more effective interventions and research to improve students' quality of life.

Lifestyle medicine is the medical practice involving several variables that influence an individual's health-disease process. It is a new area of medicine that studies and proposes to prevent and treat mainly chronic diseases and guide actions and strategies for health education. Encouraging educational programs and means to stimulate the improvement of life habits is essential for the recovery of college students with low academic performance and poor lifestyle. Based on structuring intra and intersectoral measures to support and make them co-responsible for a quality education allied to their health. We understand the need for this improvement in all areas of knowledge and for all UNILA students. However, suppose we must select a priority group among the surveyed courses, we would suggest actions directed towards the area of the Human Sciences since it had the lowest scores in both the Sense of Coherence and Lifestyle questionnaires.

## Acknowledgments

We want to thank everyone involved in this research, especially the course coordinators who opened the doors to us to apply the questionnaires and each who took the time to answer them thoughtfully. We also want to thank the students from LUMES (university league of lifestyle medicine of UNILA) Caroline Sousa da Silva and Edimar Pereira Nunes for their help in the beginning of the research.

## Author Contributions

**Conceptualization:** João Paulo Costa Braga, Eduardo Wolfgram, João Paulo Batista de Souza, Roberto de Almeida, Cezar Rangel Pestana.

**Data curation:** João Paulo Costa Braga, Eduardo Wolfgram, João Paulo Batista de Souza, Roberto de Almeida, Cezar Rangel Pestana.

**Formal analysis:** João Paulo Costa Braga, Cezar Rangel Pestana.

**Investigation:** João Paulo Costa Braga, Eduardo Wolfgram, Cezar Rangel Pestana.

**Methodology:** João Paulo Costa Braga, Eduardo Wolfgram, João Paulo Batista de Souza, Roberto de Almeida, Cezar Rangel Pestana.

**Project administration:** João Paulo Costa Braga.

**Supervision:** João Paulo Costa Braga, Roberto de Almeida, Cezar Rangel Pestana.

**Validation:** João Paulo Costa Braga, Cezar Rangel Pestana.

**Visualization:** João Paulo Costa Braga, Eduardo Wolfgram.

**Writing – original draft:** João Paulo Costa Braga, Eduardo Wolfgram, João Paulo Batista de Souza, Larissa Gabriele Fausto Silva, Yonel Estavien, Cezar Rangel Pestana.

**Writing – review & editing:** João Paulo Costa Braga, Eduardo Wolfgram, João Paulo Batista de Souza, Larissa Gabriele Fausto Silva, Yonel Estavien, Roberto de Almeida, Cezar Rangel Pestana.

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
