## [Decision Letter · Decision Letter 0]

25 May 2023

PONE-D-23-08485Lifestyle and Sense of Coherence: a comparative analysis among university students in different areas of knowledgePLOS ONE

Dear Dr. Braga,

Thank you for submitting your manuscript to PLOS ONE. After careful consideration, we feel that it has merit but does not fully meet PLOS ONE’s publication criteria as it currently stands. Therefore, we invite you to submit a revised version of the manuscript that addresses the points raised during the review process.

We look forward to receiving your revised manuscript.

Kind regards,

Maria José Nogueira, Ph.D.

Academic Editor

PLOS ONE

Journal Requirements:

5. Please ensure that you refer to Figure 1 in your text as, if accepted, production will need this reference to link the reader to the figure.

6. We note that Figure 1 in your submission contain copyrighted images. All PLOS content is published under the Creative Commons Attribution License (CC BY 4.0), which means that the manuscript, images, and Supporting Information files will be freely available online, and any third party is permitted to access, download, copy, distribute, and use these materials in any way, even commercially, with proper attribution. For more information, see our copyright guidelines: http://journals.plos.org/plosone/s/licenses-and-copyright.

(1) You may seek permission from the original copyright holder of Figure 1 to publish the content specifically under the CC BY 4.0 license. 

**Additional Editor Comments:**

Dear Author, your work has been reviewed and it is considered that minor changes are necessary.

The work is relevant and within the scope of the journal.

It is well-founded and has a robust statistical treatment.

The literature review should be improved with more recent references.

Reviewers' comments:

Reviewer's Responses to Questions

**Comments to the Author**

1. Is the manuscript technically sound, and do the data support the conclusions?

Reviewer #1: Yes

Reviewer #2: Yes

2. Has the statistical analysis been performed appropriately and rigorously? 

Reviewer #1: Yes

Reviewer #2: Yes

3. Have the authors made all data underlying the findings in their manuscript fully available?

Reviewer #1: Yes

Reviewer #2: Yes

4. Is the manuscript presented in an intelligible fashion and written in standard English?

Reviewer #1: Yes

Reviewer #2: Yes

5. Review Comments to the Author

Reviewer #1: Maybe it will be good to have some references regarding cultural validation about this subjet.

Very interesting but maybe the subjeto is very limited to the reality of a small circle.

Some references must be actualized.

Reviewer #2: Congratulations on the manuscript, pertinence, work done, and clarity.

Introduction: well structured and with the appropriate items

Method: makes reference to aspects of scientific and methodological rigor

Mention of ethical aspects

Excellent discussion of the data, however the references should be updated.

Bibliography: Update references - a great percentage are more than 10 years

6. PLOS authors have the option to publish the peer review history of their article (what does this mean?). If published, this will include your full peer review and any attached files.

Reviewer #1: **Yes: **Raul Alberto Carrilho Cordeiro

Reviewer #2: No

---

## [Author Response · Author response to Decision Letter 0]

24 Jun 2023

Dear Dr. Maria José

A revised version of the manuscript is presented with changes in red, italic, between Parenthesis ( ), and below each reviewer comment. 

The following points were addressed:

• A rebuttal letter uploaded as a separate file labeled 'Response to Reviewers'.

• A marked-up copy of the manuscript uploaded as a separate file labeled 'Revised Manuscript with Track Changes'.

• An unmarked version of the revised paper without tracked changes, uploaded as a separate file labeled 'Manuscript'.

Journal Requirements:

(Figure citations are in accordance with file names as shown in the formatting guidelines.)

(All data used to support the results are fully available at https://datadryad.org/stash/share/_RlHZJP5BJckyC4EZQ2_Q-Rjtj39aMxSbe-3sLTDhko and https://doi.org/10.5061/dryad.bcc2fqzhd)

(The changes included in the new version are described in the cover letter.)

(We moved Ethics statement from “Introduction” to “Material and methods” section.)

5. Please ensure that you refer to Figure 1 in your text as, if accepted, production will need this reference to link the reader to the figure.

(We have not obtained permission from the copyright holder, so figure 1 was removed from the manuscript.)

6. We note that Figure 1 in your submission contain copyrighted images. All PLOS content is published under the Creative Commons Attribution License (CC BY 4.0), which means that the manuscript, images, and Supporting Information files will be freely available online, and any third party is permitted to access, download, copy, distribute, and use these materials in any way, even commercially, with proper attribution. For more information, see our copyright guidelines: http://journals.plos.org/plosone/s/licenses-and-copyright.

(1) You may seek permission from the original copyright holder of Figure 1 to publish the content specifically under the CC BY 4.0 license. 

(We removed the figure.)

(All references were checked. The new version also includes a general reference updating.

References

- Updated but keeping the same author: 1, 3 and 17.

- Checked: 2, 4,6,7, 8,11, 12, 13, 14, 16, 18, 20, 21, 22, 24, 25, 26, 29, 31,32,33, 34, 35, 36, 38, 39, 41, 42, 43, 45, 46 and 47.

- Deleted and replaced with more updated ones: 5,11 and 15.

- Added: 9,10,15, 19,23, 27, 28, 30, 37, 40, 44, 48 and 49.)

8. Review Comments to the Author

Reviewer #1: Maybe it will be good to have some references regarding cultural validation about this subject.

(We have included the most important studies conducted in Brazil in different social context of Lifestyle and Sense of Coherence. The following reference numbers are directly associated with these approach: 6,9,10, 12, 13, 14, 16, 24, 29, 32, 33, 34, 41 and 45. These references provide valuable insights into the cultural context to the overall validation of our research.)

Very interesting but maybe the subject is very limited to the reality of a small circle.

Some references must be actualized.

(We would like to point out that family medicine and health community is more particularly engaged with domains covered by FANTASTICO questionnaire. Despite the concept of salutogenesis and sense of coherence may not be commonly used in medical intervention, these aspects have been increasingly emphasized as modifiable risk factors to many diseases and how people are committed to heathcare.

We hope to add to the body of knowledge by examining these domains and offer new insights to healthcare practice guidelines.)

Reviewer #2: Congratulations on the manuscript, pertinence, work done, and clarity.

Introduction: well structured and with the appropriate items

Method: makes reference to aspects of scientific and methodological rigor

Mention of ethical aspects

Excellent discussion of the data, however the references should be updated.

Bibliography: Update references - a great percentage are more than 10 years

(We have now included more recent and significant studies described in the literature with more than 60% of references published in the last six years. These references are in line with more recent understanding about the current state of research in the field. We believe this update considerably improved the quality of our research.)

Additional Editor Comments:

Dear Author, your work has been reviewed and it is considered that minor changes are necessary.

The work is relevant and within the scope of the journal.

It is well-founded and has a robust statistical treatment.

The literature review should be improved with more recent references.

Reviewers' comments:

Reviewer's Responses to Questions

Comments to the Author

1. Is the manuscript technically sound, and do the data support the conclusions?

Reviewer #1: Yes

Reviewer #2: Yes

2. Has the statistical analysis been performed appropriately and rigorously?

Reviewer #1: Yes

Reviewer #2: Yes

3. Have the authors made all data underlying the findings in their manuscript fully available?

Reviewer #1: Yes

Reviewer #2: Yes

4. Is the manuscript presented in an intelligible fashion and written in standard English?

Reviewer #1: Yes

Reviewer #2: Yes

5. Review Comments to the Author

Reviewer #1: Maybe it will be good to have some references regarding cultural validation about this subjet.

Very interesting but maybe the subjeto is very limited to the reality of a small circle.

Some references must be actualized.

Reviewer #2: Congratulations on the manuscript, pertinence, work done, and clarity.

Introduction: well structured and with the appropriate items

Method: makes reference to aspects of scientific and methodological rigor

Mention of ethical aspects

Excellent discussion of the data, however the references should be updated.

Bibliography: Update references - a great percentage are more than 10 years

6. PLOS authors have the option to publish the peer review history of their article (what does this mean?). If published, this will include your full peer review and any attached files.

Do you want your identity to be public for this peer review? For information about this choice, including consent withdrawal, please see our Privacy Policy.

Reviewer #1: Yes: Raul Alberto Carrilho Cordeiro

Reviewer #2: No

(We uploaded the figure files to PACE digital diagnostic tool, in order to ensure that figures meet PLOS requirements.)

---

## [Editor Report · Decision Letter 1]

2 Jul 2023

Lifestyle and Sense of Coherence: a comparative analysis among university students in different areas of knowledge

PONE-D-23-08485R1

Dear Dr. Braga,

We’re pleased to inform you that your manuscript has been judged scientifically suitable for publication and will be formally accepted for publication once it meets all outstanding technical requirements.

Kind regards,

Maria José Nogueira, Ph.D.

Academic Editor

PLOS ONE
---

## [Editor Report · Acceptance letter]

12 Jul 2023

PONE-D-23-08485R1 

Lifestyle and Sense of Coherence: a comparative analysis among university students in different areas of knowledge 

Dear Dr. Braga:

I'm pleased to inform you that your manuscript has been deemed suitable for publication in PLOS ONE. Congratulations! Your manuscript is now with our production department. 

Kind regards, 

on behalf of

Professor Maria José Nogueira 

Academic Editor

PLOS ONE